# Repurposing Hsp90 inhibitors as antimicrobials targeting two-component systems identifies compounds leading to loss of bacterial membrane integrity

Blanca Fernandez-Ciruelos,[1] Marco Albanese,[2,3] Anmol Adhav,[4] Vitalii Solomin,[5] Arabela Ritchie-Martinez,[1] Femke Taverne,[1] Nadya Velikova,[1] Aigars Jirgensons,[5] Alberto Marina,[4] Paul W. Finn,[2,3] Jerry M. Wells[1]

**ABSTRACT**   The discovery of antimicrobials with novel mechanisms of action is crucial to tackle the foreseen global health crisis due to antimicrobial resistance. Bacterial two-component signaling systems (TCSs) are attractive targets for the discovery of novel antibacterial agents. TCS-encoding genes are found in all bacterial genomes and typically consist of a sensor histidine kinase (HK) and a response regulator. Due to the conserved Bergerat fold in the ATP-binding domain of the TCS HK and the human chaperone Hsp90, there has been much interest in repurposing inhibitors of Hsp90 as antibacterial compounds. In this study, we explore the chemical space of the known Hsp90 inhibitor scaffold 3,4-diphenylpyrazole (DPP), building on previous literature to further understand their potential for HK inhibition. Six DPP analogs inhibited HK autophosphorylation *in vitro* and had good antimicrobial activity against Gram-positive bacteria. However, mechanistic studies showed that their antimicrobial activity was related to damage of bacterial membranes. In addition, DPP analogs were cytotoxic to human embryonic kidney cell lines and induced the cell arrest phenotype shown for other Hsp90 inhibitors. We conclude that these DPP structures can be further optimized as specific disruptors of bacterial membranes providing binding to Hsp90 and cytotoxicity are lowered. Moreover, the X-ray crystal structure of resorcinol, a substructure of the DPP derivatives, bound to the HK CheA represents a promising starting point for the fragment-based design of novel HK inhibitors.

**IMPORTANCE** The discovery of novel antimicrobials is of paramount importance in tackling the imminent global health crisis of antimicrobial resistance. The discovery of novel antimicrobials with novel mechanisms of actions, e.g., targeting bacterial two-component signaling systems, is crucial to bypass existing resistance mechanisms and stimulate pharmaceutical innovations. Here, we explore the possible repurposing of compounds developed in cancer research as inhibitors of two-component systems and investigate their off-target effects such as bacterial membrane disruption and toxicity. These results highlight compounds that are promising for further development of novel bacterial membrane disruptors and two-component system inhibitors.

**KEYWORDS**   antimicrobial, repurposing, Hsp90, two-component system, membrane disruption

The rise of antibiotic resistance worldwide makes it necessary to find new antimicrobial treatments, preferably with low potential for resistance development (1). Two-component systems (TCSs), the most important signaling systems in bacteria, are promising antibacterial targets (2). They are absent in animal cells (3) and present in all bacteria (4). Each bacterial species encodes multiple TCSs (5–7) involved in adaptive

Address correspondence to Blanca Fernandez-Ciruelos, blanca.maria.fdez.ciruelos@gmail.com, or Jerry M. Wells, Jerry.wells@wur.nl.

Blanca Fernandez-Ciruelos and Marco Albanese contributed equally to this article. Name order was decided on the basis of writing contribution.

The authors declare no conflict of interest.

See the funding table on p. 17.

responses and the regulation of metabolism (8), response to extracellular stresses (9), antibacterial resistance (10–13), and virulence in the host (14–17), among others. Typically, TCSs consist of a sensor histidine kinase (HK) and a response regulator (RR) (18). Prototypical HKs are membrane-bound dimers comprising a sensor domain that detects their cognate stimuli, a catalytic ATP binding domain (CA) that binds ATP and, upon signal detection, autophosphorylates a conserved histidine in the dimerization, and a histidine phosphotransfer domain (DHp). Subsequently, the phosphate is shuttled to a conserved aspartic acid in the RR, typically leading to dimerization and high affinity binding to regulatory motifs in the genome (19–21). The CA and DHp domains are well conserved in HKs within and among different bacteria (22, 23), making them good target sites for the simultaneous inhibition of multiple TCSs and the design of broad-spectrum agents. This polypharmacological approach is expected to limit the emergence of target-based resistance mechanisms (24). Due to the essentiality of some TCSs [e.g., WalKR in *Staphylococcus aureus* (25) and *Bacillus subtilis* (26)] and their role in adaptation, it has been proposed that inhibition of multiple TCSs would compromise growth and/or attenuate survival and virulence in the host (27, 28).

One approach for the inhibition of multiple TCSs has been the design of ATP-competitive inhibitors of HKs, with the aim of inhibiting autophosphorylation and TCS signaling (27). The ATP-binding pocket of HKs adopts the Bergerat fold, an α/β sandwich characterized by four conserved regions (the N-box, the G1-box, the G2-box, and the G3-box) and the highly variable ATP-lid (29). The Bergerat fold has been observed in the ATP-binding domain of the GHKL protein superfamily, which includes DNA gyrases, the molecular chaperone Hsp90, bacterial HKs, and the MutL mismatch repair enzyme (29). Due to this similarity, ATP-competitive Hsp90 inhibitors were proposed as hits for HK inhibition (30). Multiple inhibitors targeting the ATP binding pocket of Hsp90 have been described (31, 32) due to its attractiveness as a target for anti-cancer drugs (33), and some have been shown to weakly bind to the HK PhoQ from *Salmonella* (30). Vo et al. (34) showed that among six well-established Hsp90 inhibitors, CCT018159 (35), a 3,4-diphenylpyrazole (DPP) (Fig. 1A), was the most potent HK inhibitor of CckA from *Caulobacter crescentus* (IC$_{50}$ = 30 µM) and PhoQ from *Salmonella typhimurium* (IC$_{50}$ = 261 µM). Structure-activity relationship (SAR) investigation around the DPP scaffold led to marginal improvements in HK inhibition, with compound **5b** representing the most potent inhibitor of the series (Fig. 1A, IC$_{50}$ CckA = 14 µM; IC$_{50}$ PhoQ = 238 µM). Compound **5b** also showed moderate antibacterial activity against *Escherichia coli* DC2 (a hypersensitive *E. coli* strain), *C. crescentus*, and *B. subtilis* (MIC range 12–74 µg/mL) (34).

DPPs are anchored to the ATP-binding site of Hsp90 through a network of hydrogen bonds established by the resorcinol and the pyrazole systems (rings A and B, respectively; Fig. 1A). As illustrated by the CCT018159-Hsp90α crystal complex in Fig. 1B [PDB 2BT0 (36)], the 1-hydroxyl from the resorcinol group forms a direct hydrogen bond with the side chain of a buried aspartate residue (Asp93) while the 3-hydroxyl establishes water-mediated interactions. The pyrazole's N2 forms a water-bridged hydrogen bond with Asp93 and the backbone NH of Gly97. The residues Asn51, Asp93, and Gly97, together with the ordered water molecules (W1, W2, and W3), are highly conserved in HKs as they are involved in binding the adenine system of ATP (Fig. 1C). Hence, interactions with these residues are likely to be preserved when repurposing Hsp90 inhibitors as HK inhibitors. The potential for off-target cytotoxic effects by inhibition of mammalian Hsp90 has to be taken into account when using the repurposing approach.

In a previous study, we have reported synthetic derivatives featuring a DPP core with antibacterial activity against *S. aureus* (39). Herein, we investigate the antibacterial activity and potential inhibition of HKs for a subset of these derivatives and related compounds from commercial vendors as well as the mechanisms responsible for their antibacterial activity. By studying the possible off-target effects and cytotoxicity in mammalian cells, we showed that DPPs were inhibiting bacterial growth by interfering with membrane integrity while retaining cytotoxicity to mammalian cells via different mechanisms, membrane damage, and Hsp90 inhibition. This highlights the challenges of

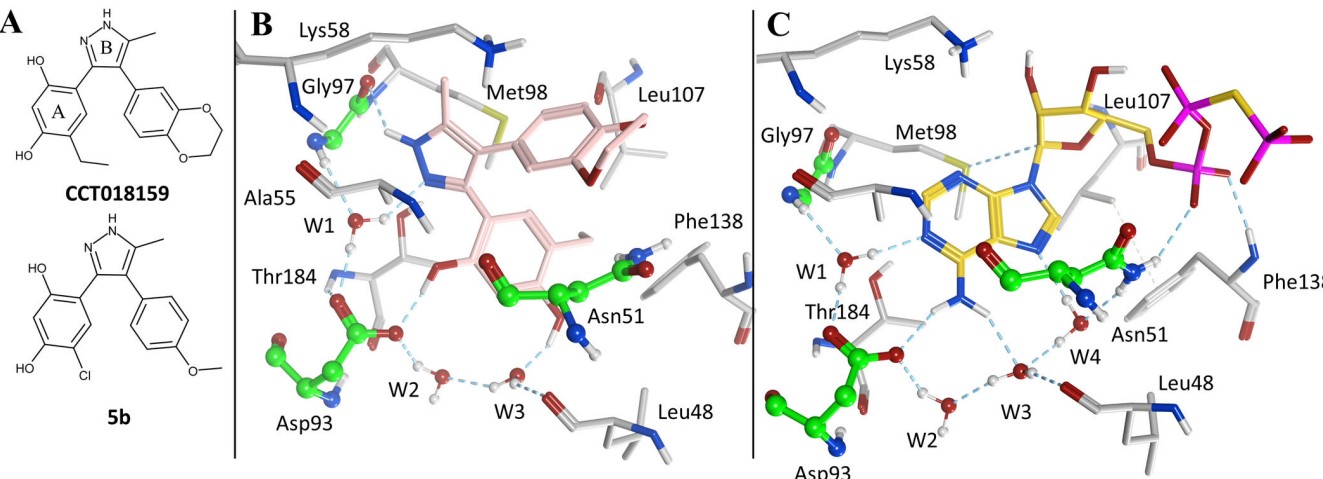

**FIG 1** Reported DPP-based HK inhibitors and experimental binding mode to Hsp90α. (A) 2D chemical structure of representative HK inhibitors with the DPP scaffold that includes the resorcinol (ring A) and pyrazole systems (ring B). (B) Crystal binding mode of the DPP compound CCT018159 (pink carbon atoms) in complex with Hsp90α [PDB 2BT0 (36)]. (C) Crystal binding mode of a non-hydrolyzable ATP analog (phosphomethylphosphonic acid adenylate ester; yellow carbon atoms) in complex with Hsp90α [PDB 3T10 (37)]. The carbon atoms of residues highly conserved in HKs are depicted in green while remaining residues in contact with the ligand are colored in gray. Hydrogen bonds are shown as dashed lines. Image generated in Molecular Operating Environment (MOE) (38).

repurposing Hsp90 inhibitors as antimicrobials targeting TCS HKs but indicates possibilities to exploit DPP analogs as agents specifically targeting bacterial membranes. Finally, the X-ray crystal structure of resorcinol bound to HK CheA looked attractive for the fragment-based design of novel HK inhibitors.

## RESULTS

### Structure-activity relationship of antimicrobial 3,4-diphenylpyrazole compounds against *S. aureus*

A collection of 24 DPP analogs was synthesized (39) or purchased to further expand the SAR around the diphenylpyrazole scaffold (Fig. 2). These analogs were first tested for minimal inhibitory concentration against the Gram-positive *S. aureus*.

With the exception of **DPP-23**, all the compounds inhibited *S. aureus* growth with MIC between 1.56 and 50 µg/mL (Fig. 2). Differently from the parent Hsp90 inhibitor series, several analogs tested are O-alkylated at the $R_2$ positions of ring A and the antibacterial potency of the methoxy and ethoxy analogs is generally comparable to the hydroxyl derivatives, while benzyloxy is among the most potent compounds. Interestingly, removing one (**DPP-22**) or both hydroxyl groups (**DPP-21**) of the resorcinol system results in highly active compounds.

In ring B, isoxazole acts as a pyrazole bioisostere as it does not alter the MIC (compare **DPP-5** with **DPP-11**). Replacing $R_1$ methyl with a trifluoromethyl group (see pairs **DPP-5/DPP-9** and **DPP-6/DPP-10**) increases antibacterial potency. Removal of ring C is well tolerated in the $R_2$-methoxy derivative **DPP-24** but significantly decreases the MIC in the resorcinol analog **DPP-23**. When ring C is present, the $R_4$-chloro derivatives are more active against *S. aureus* than matched compounds lacking the chlorine substituent.

All tested compounds were toxic to human embryonic kidney (HEK293) cells (Fig. 2). However, only a moderate correlation was found between MIC and toxicity (Pearson correlation $r = 0.4289$, $P$ value = 0.0411; Fig. SF1 A), indicating that mechanisms of antimicrobial activity and toxicity may be different.

Finally, when compounds were tested in the presence of 20% FCS, the MIC of all tested compounds was >250 µg/mL, indicating that these compounds bind to serum proteins. This is expected due to the high cLogP of the structures (cLogP 3–4).

| DPP comp. | X | $R_1$ | $R_2$ | $R_3$ | $R_4$ | $R_5$ | MIC *S. aureus* μg/ml (μM) | LC$_{50}$ HEK293 cells μg/ml (μM) |
|---|---|---|---|---|---|---|---|---|
| 1 | N | -CF$_3$ | -OH | -H | -H | -H | 6.25 (19.53) | 4.06 (12.69) |
| 2 | N | -CF$_3$ | -OMe | -H | -H | -H | 12.50 (37.3)† | 7.34 (21.91) |
| 3 | N | -CF$_3$ | -OEt | -H | -H | -H | 6.25 (17.86) | 25.64 (76.53) |
| 4 | N | -CF$_3$ | -OBn | -H | -H | -H | 1.56 (3.80)† | 7.26 (17.71) |
| 5 | N | -CF$_3$ | -OH | -H | -Cl | -H | 3.12 (8.79) | 12.88 (36.28) |
| 6 | N | -CF$_3$ | -OMe | -H | -Cl | -H | 3.12 (8.43)† | 14.30 (38.65) |
| 7 | N | -CF$_3$ | -OEt | -H | -Cl | -H | 3.12 (8.21) | 10.82 (28.47) |
| 8 | N | -CF$_3$ | -OBn | -H | -Cl | -H | 1.56 (3.51)† | 14.25 (32) |
| 9 | N | -Me | -OH | -H | -Cl | -H | 25 (83.33)† | 12.30 (41) |
| 10 | N | -Me | -OMe | -H | -Cl | -H | 25 (79.37) | 23.23 (73.74) |
| 11 | O | -CF$_3$ | -OH | -H | -Cl | -H | 3.12 (8.76)† | 3.41 (9.61) |
| 12 | N | -CF$_3$ | -OH | -H | -OMe | -H | 12.50 (35.71) | 31.74 (90.69) |
| 13 | N | -CF$_3$ | -OMe | -H | -OMe | -H | 12.50 (34.34)† | 29.88 (82.09) |
| 14 | N | -CF$_3$ | -OEt | -H | -OMe | -H | 6.25 (16.45) | 6.57 (17.29) |
| 15 | N | -CF$_3$ | -OBn | -H | -OMe | -H | 3.12 (7.09) | 9.64 (21.91) |
| 16 | N | -CF$_3$ | -OH | -OMe | -H | -H | 25 (71.43) | 16.35 (46.71) |
| 17 | N | -CF$_3$ | -OMe | -OMe | -H | -H | 25 (68.49) | 12.58 (34.47) |
| 18 | N | -CF$_3$ | -OH | -H | -OMe | -OMe | 50 (131.58) | 22.81 (60.03) |
| 19 | N | -CF$_3$ | -OMe | -H | -OMe | -OMe | 25 (63.29) | 13.65 (34.56) |
| 20 | N | -CF$_3$ | -OEt | -H | -OMe | -OMe | 12.5 (30.49) | 16.53 (40.32) |
| 21* | N | -CF$_3$ | -H | -H | -Cl | -H | 1.56 (4.8)† | 7.12 (21.91) |
| 22 | N | -CF$_3$ | -H | -H | -Cl | -H | 1.56 (4.59) | 2.89 (8.5) |
| 23** | N | -CF$_3$ | -OH | -H | -H | -H | 250 (961.54)† | >250 (>650) |
| 24** | N | -CF$_3$ | -OMe | -H | -H | -H | 25 (102.04)† | 6.79 (27.71) |

*1-OH is absent ** C-ring is absent † Data previously published in Solomin *et al*(39).

**FIG 2** Structure, activity against *S. aureus*, and toxicity in HEK cells of a series of diphenylpyrazole compounds. Activity against *S. aureus* is expressed as minimal inhibitory concentration in micrograms per milliliter. Toxicity in HEK cells is expressed as lethal concentration 50 (LC$_{50}$) - the concentration required to kill 50% of the cell population. Compound substitutions ($R_1$ to $R_5$ and X) from the A, B, or C rings of the scaffold as indicated in the diagram are also depicted.

## Spectrum of activity

To determine the potential of the DPP compounds as broad-spectrum antibiotics, we tested their antibacterial activity against the Gram-positive organisms *Enterococcus faecium* and *Enterococcus faecalis*, as well as the Gram-negatives *E. coli*, *Pseudomonas aeruginosa*, and *Pasteurella haemolytica*, which are all important human or animal pathogens (Table 1 and Table ST1). The range of activity of the compounds is similar in all Gram-positive bacteria. The tested compounds show no activity against Gram-negative *E. coli* and *P. aeruginosa* and only moderate activity against *P. haemolytica*.

We also tested the inhibitory activity of compounds in two different *E. coli* mutants: (i) *E. coli* JW5503 that lacks TolC efflux pumps and (ii) *E. coli* D21f2 that has a defective LPS inner core increasing permeability of the outer membrane. *E. coli* D21f2 has a slightly higher susceptibility to DPP compounds than *E. coli* ATCC25922. MIC data for *E. coli* JW5503 are similar to those obtained for Gram-positive bacteria, indicating that TolC-dependent efflux plays a major role in the susceptibility of *E. coli* to the DPP inhibitors. Table 1 shows MIC data of compounds that show inhibition of histidine kinases (see next section) against the panel of strains.

## Docking and inhibition studies

Docking studies with the HK PhoQ from *E. coli* suggest that only a subset of the DPP compounds can adopt a binding mode similar to the CCT018159-Hsp90α complex, exemplified by the putative binding pose of **DPP-5** (Fig. 3A and B). In detail, the five-membered ring of **DPP-5** forms water-mediated contacts with Asp415 and Gly419, while Ile420 and Tyr393 flank the two faces of the pyrazole. Asp415 and Gly419 are part of the G1-box and are highly conserved among members of the GHKL family (Fig. 3C). On the contrary, aromatic residues at the position corresponding to the Tyr393 position are HK specific (this residue corresponds to Ala55 in Hsp90α; Fig. 3C) (29). Ring C and the trifluoromethyl group are projected toward the opening of the ATP-binding pocket. The 1-hydroxyl of the resorcinol scaffold engages the conserved aspartate (Asp415), while the 3-hydroxyl binds the carbonyl backbone of Val386. However, by analogy with the DPP binding mode to Hsp90 shown in Fig. 1, the 3-hydroxyl could also participate in the hydrogen bond network with a conserved water molecule system (W2-W3). The docking settings did not include the water molecules W2 and W3 to allow for their potential displacement when the 3-hydroxyl is alkylated. Due to the limited size of the subpocket housing these water molecules (shown as a surface in Fig. 3), derivatives with bulky substituents in position 3, such as the benzyloxy group of **DPP-8**, are unlikely to retain this binding mode unless there are major protein conformational changes. The *in silico* studies predict a flipped binding mode for **DPP-8**, in which the pyrazole nitrogen atoms are hydrogen bonded to Asp415 and W1 and the trifluoromethyl group displaces the water network. Ring A is flanked by Gly419, Ile420, and Tyr393, with the benzyloxy group partially solvent exposed, establishing contacts with Pro418 and Pro421. Ring C faces the hydrophobic floor of the pocket formed by the side chains of Ile428, Leu446, and Met472. An analogous binding pose is also predicted for the derivative lacking both hydroxyl groups (**DPP-22**).

**TABLE 1** MIC data for DPP compounds that showed inhibition of HKs tested against a panel of Gram-positive and Gram-negative strains, including *E. coli* outer membrane and efflux mutants

| DPP comp. | | | MIC (µg/mL) | | | | |
| --- | --- | --- | --- | --- | --- | --- | --- |
| | *E. faecium* | *E. faecalis* | *P. haemolytica* | *P. aeruginosa* | *E. coli* | *E. coli* JW5503 | *E. coli* D21f2 |
| 2 | 25 | 25 | 50 | >250 | >250 | 6.25 | 50 |
| 5 | 12.5 | 12.5 | 12.5 | >250 | 50 | 3.12 | 50 |
| 6 | 6.25 | 6.25 | 12.5 | >250 | >250 | 3.12 | 25 |
| 7 | 3.12 | 3.12 | 25 | >250 | >250 | 3.12 | 25 |
| 12 | 50 | 50 | 50 | >250 | >250 | 12.5 | >250 |
| 13 | 25 | 25 | >250 | >250 | >250 | 12.5 | >250 |

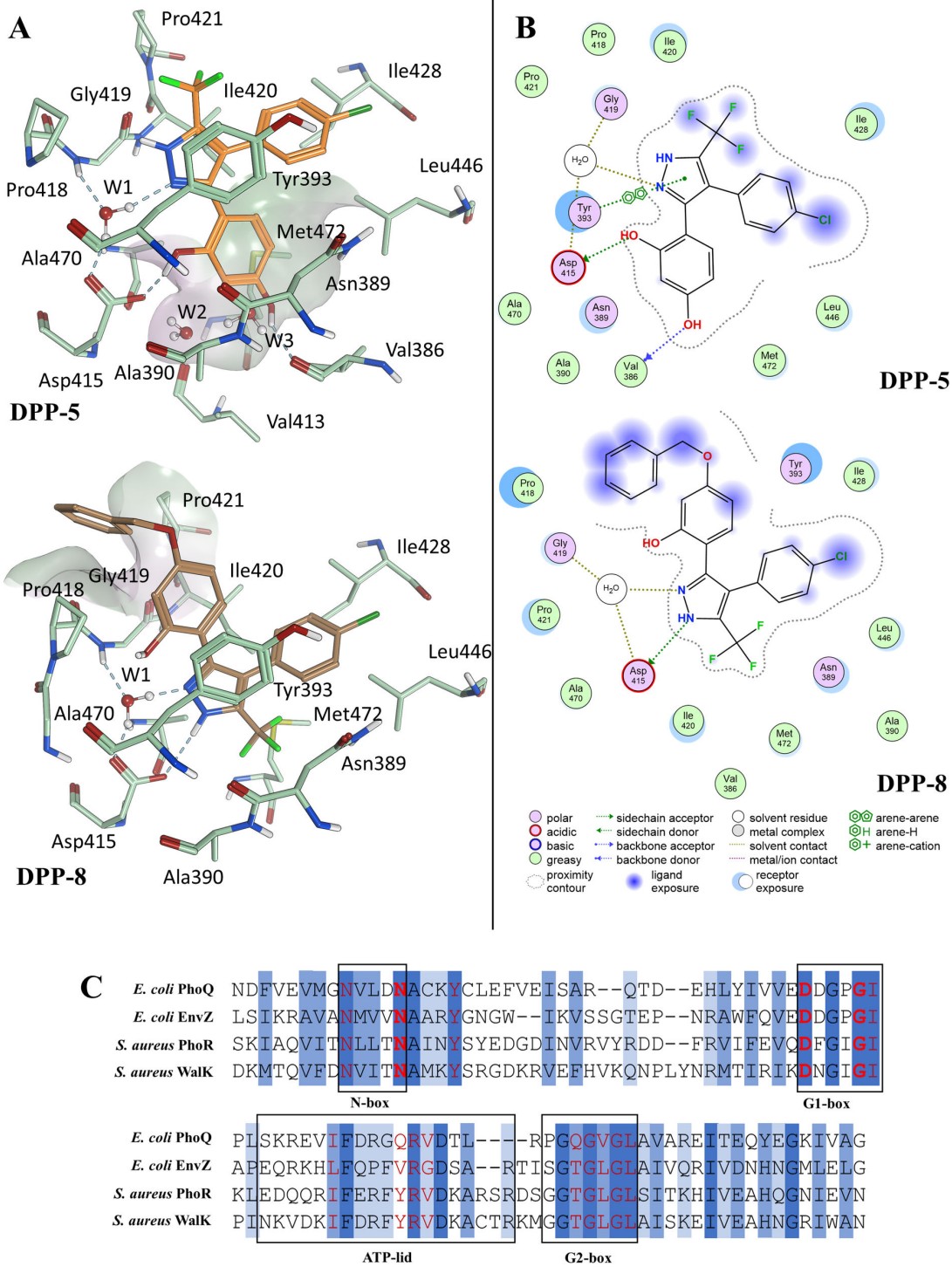

**FIG 3** *In silico* binding modes of representative DPP compounds and HK sequence alignment. (A) 3D models of the top-ranked GOLD docking pose of DPP-5 and DPP-8 in complex with the HK PhoQ from *E. coli* (PDB 1ID0). A surface delineates: the subpocket housing W2-W3 in the DPP-5 pose; the putative interaction site of the benzyloxy of DPP-8 (surface color code: purple = hydrophobic; white = neutral; green = lipophilic). Image prepared using MOE. (B) MOE ligand interaction diagrams of the top-ranked GOLD docking pose of DPP-5 and DPP-8 in complex with the HK PhoQ from *E. coli* (PDB 1ID0). (C) Protein sequence alignment of *E. coli* PhoQ, *E. coli* EnvZ, *S. aureus* PhoR, and *S. aureus* WalK. Amino-acids in red are predicted to interact with ATP. In red and bold is the conserved triad (N389, D415, and G419 in *E. coli* PhoQ); Y393 is conserved in HKs but is absent in Hsp90 (dark-blue, 100% amino acid conservation; blue, amino acid physicochemical property conserved; light-blue, amino acid conserved in at least 80% of the HKs).

The compounds were tested for inhibition of HK autophosphorylation *in vitro*. Of the 24 derivatives, 6 inhibited *in vitro* autophosphorylation of *E. coli* EnvZ at a concentration of 2 mM. For four compounds that showed total inhibition in the initial screening, the IC$_{50}$ (amount of compound required to reduce HK autophosphorylation by 50%) was measured and was in the high micromolar range for *S. aureus* PhoR and *E. coli* EnvZ (Table 2; Fig. SF2). Each compound has similar IC$_{50}$ values for PhoR and EnvZ, indicating good potential for polypharmacology. We discounted the notion that DPP-induced aggregation of HKs was the reason for inhibiting autophosphorylation using native gel protein electrophoresis of HK proteins with and without DPPs (data not shown). We also investigated if the four DPP inhibiting HK autophosphorylation would also bind to Hsp90α in a fluorescence polarization assay. In this assay, inhibitors were tested for their ability to displace FITC-labeled geldanamycin (a well-known Hsp90 inhibitor) from the ATP pocket (40). All the compounds tested displaced FITC-labeled geldanamycin with IC$_{50}$s in the nM range (Table 2).

We also tested whether the DPP series would inhibit the bacterial gyrase, which also possesses the Bergerat fold in their ATP-binding domain (29), as this would lead to antimicrobial activity. None of the DPP compounds inhibited gyrase activity, indicating this mechanism of action is not responsible for their antimicrobial activity (Fig. SF3).

## Resorcinol fragment binds to the ATP-binding site of the histidine kinase CheA

Our attempts to co-crystallize the DPP derivative with the CA domain of CheA from *T. maritima*, a class II HK for which robust crystallization protocols have been established, failed. However, we successfully obtained the X-ray crystal structure of the resorcinol fragment bound to CheA-CA with a resolution of 2.1 Å. The binding mode of the resorcinol resembles the predicted binding mode of **DPP-5** (Fig. 3) and is nearly identical to the interaction formed by the DPP CCT018159 with Hsp90α (Fig. 1). The overall structure showed a solvent-accessible binding pocket where the resorcinol was observed making contact with the conserved aspartate (Asp449) (Fig. 4A and B): the one hydroxyl from the resorcinol is in the proximity of the conserved aspartate (Asp449) and the ordered water molecule W1, while the other hydroxyl of the resorcinol forms water-bridged contacts with the backbone carbonyl a leucine residue (Leu406; Val386 in PhoQ *E. coli*). The crystal binding mode of the resorcinol fragment supports the hypothesis that resorcinol-containing DPP compounds are likely to retain the binding mode observed in Hsp90.

## DPPs increase membrane permeability in *S. aureus*

To study the effect of DPP compounds on the cytoplasmic membrane of *S. aureus*, we used the nucleic acid stain SYTOX Green. DPPs were tested for loss of membrane integrity after 5 min of exposure to 0.5×, 1×, and 2× MIC. The tested DPP compounds

**TABLE 2** *In vitro* inhibition of HK autophosphorylation by DPP compounds and Hsp90 fluorescence polarization assay results. Six DPPs showed inhibition of EnvZ autophosphorylation *in vitro*[a]

| Compound | IC$_{50}$ PhoR | IC$_{50}$ EnvZ | IC$_{50}$ Hsp90 (µM) |
| --- | --- | --- | --- |
| | *S. aureus* (µM) | *E. coli* (µM) | |
| DPP-2 | n.d. | <2mM | n.d. |
| DPP-5 | 95 ± 5.6 | 117 ± 6.3 | 0.496 |
| DPP-6 | 55 ± 3.8 | 98 ± 11.84 | 0.798 |
| DPP-7 | n.d. | <2mM | n.d. |
| DPP-12 | 328 ± 4.1 | 427 ± 9.1 | 0.574 |
| DPP-13 | 89 ± 6.9 | 175 ± 10.1 | 0.838 |

[a]The IC$_{50}$± SD (concentration of inhibitor that reduces HK autophosphorylation by 50% ± standard deviation) was calculated for five of these DPPs using *S. aureus* PhoR and *E. coli* EnvZ (*n* = 2). The results of the Hsp90 fluorescence polarization assay are reported as IC$_{50}$s (compound concentration that decreases geldanamycin binding by 50%). n.d. not done.

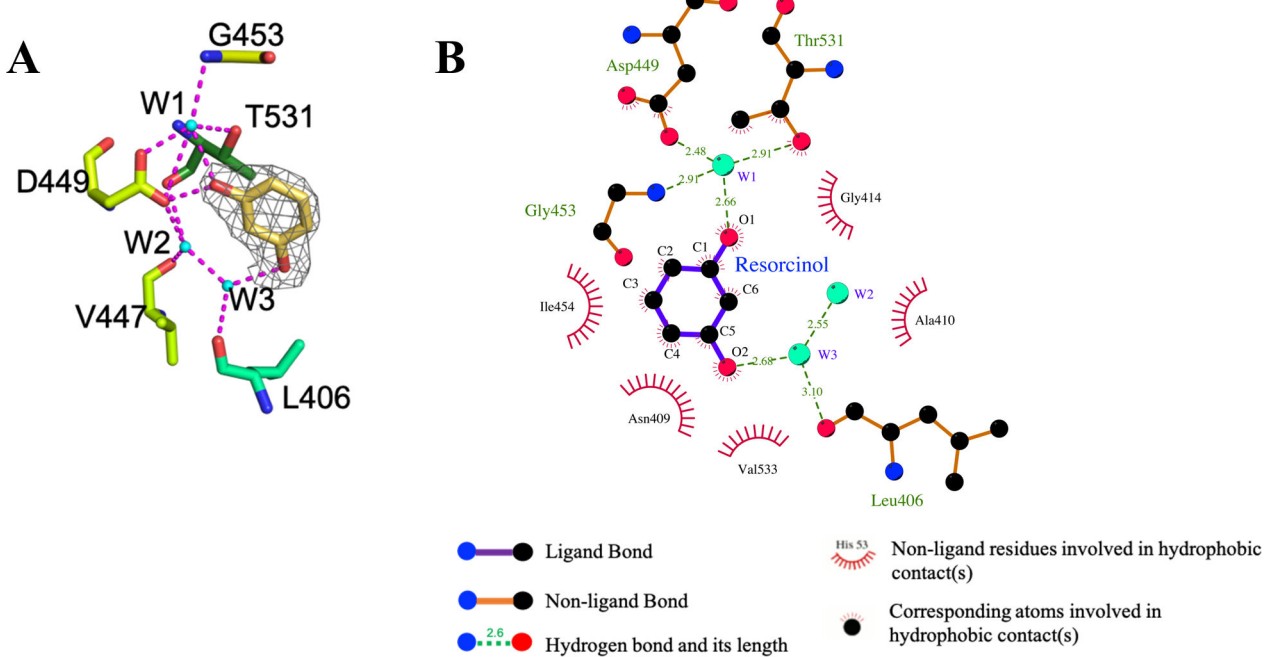

**FIG 4** Resorcinol binding mode to the binding pocket of the CheA-CA domain. (A) Fo-Fc map shows the electron density around the resorcinol (resolution 2.1 Å). Image prepared using PyMol. (B) Binding mode prediction of resorcinol to CheA-CA domain, created in LigPlot$^+$ (41). Resorcinol makes direct and indirect contacts through water molecules (W1 and W2) with the conserved aspartate (D449) residue of CheA.

significantly increased membrane permeability, similar to nisin which was used as positive control (Fig. 5; Table ST2). In contrast, no effect on membrane integrity was observed with the gyrase inhibitor novobiocin at 2× MIC. **DPP-4**, **DPP-5**, and **DPP-15** were the only compounds that did not cause significant loss of membrane integrity at 1× MIC. It could be that bulky substitutions in the $R_2$ position of ring A (**DPP-4** and **DPP-15**) reduce the ability of the compounds to cause membrane damage; however, this reduction is barely observed in **DPP-8**.

To see if the membrane-destabilizing activity of DPP compounds was specific for bacterial membranes, we tested their hemolytic activity with sheep red blood cells (RBCs) (Fig. 6). There was no correlation between the percentage of RBC hemolysis and the membrane damage caused in *S. aureus* (Pearson correlation $r = 0.3484$, $P$ value = 0.1214; Fig. SF1 B). With respect to the SAR, we see that O-methylation at position $R_4$ reduces hemolytic properties when compared with hydroxyl or chlorine derivatives (i.e., compare **DPP-2/DPP-6/DPP-13** with **DPP-3/DPP-7/DPP-14**). Hemolysis is also reduced by the substitution of the $R_1$ trifluoromethyl with a methyl group (see pairs **DPP-9/DPP-5** and **DPP-10/DPP-6**). An overview of all biological parameters per compound is included in Table ST3.

## Cytotoxicity of DPPs

The cytotoxicity of DPP compounds was quantified using the Alamar Blue assay and the human embryonic kidney cell line HEK293. All compounds with MIC ≤ 50 µg/mL against *S. aureus* were cytotoxic, with $LC_{50}$ values ranging from 2.89 to 31.7 µg/mL (Fig. 2). Eight out of 20 DPPs tested caused significant hemolysis of sheep RBCs at 250 µM after 30 min incubation (Fig. 6). We did not see a significant correlation between hemolysis and cytotoxicity $LC_{50}$s (Pearson correlation $r = -0.283$, $P$ value = 0.08; Fig. SF1 C). Since we confirmed that the DPP compounds bind to Hsp90, we hypothesized that there may be two different mechanisms of cytotoxicity, with some DPP analogs inducing membrane-associated damage to mammalian cells (such as **DPP-6**) and others Hsp90 inhibition.

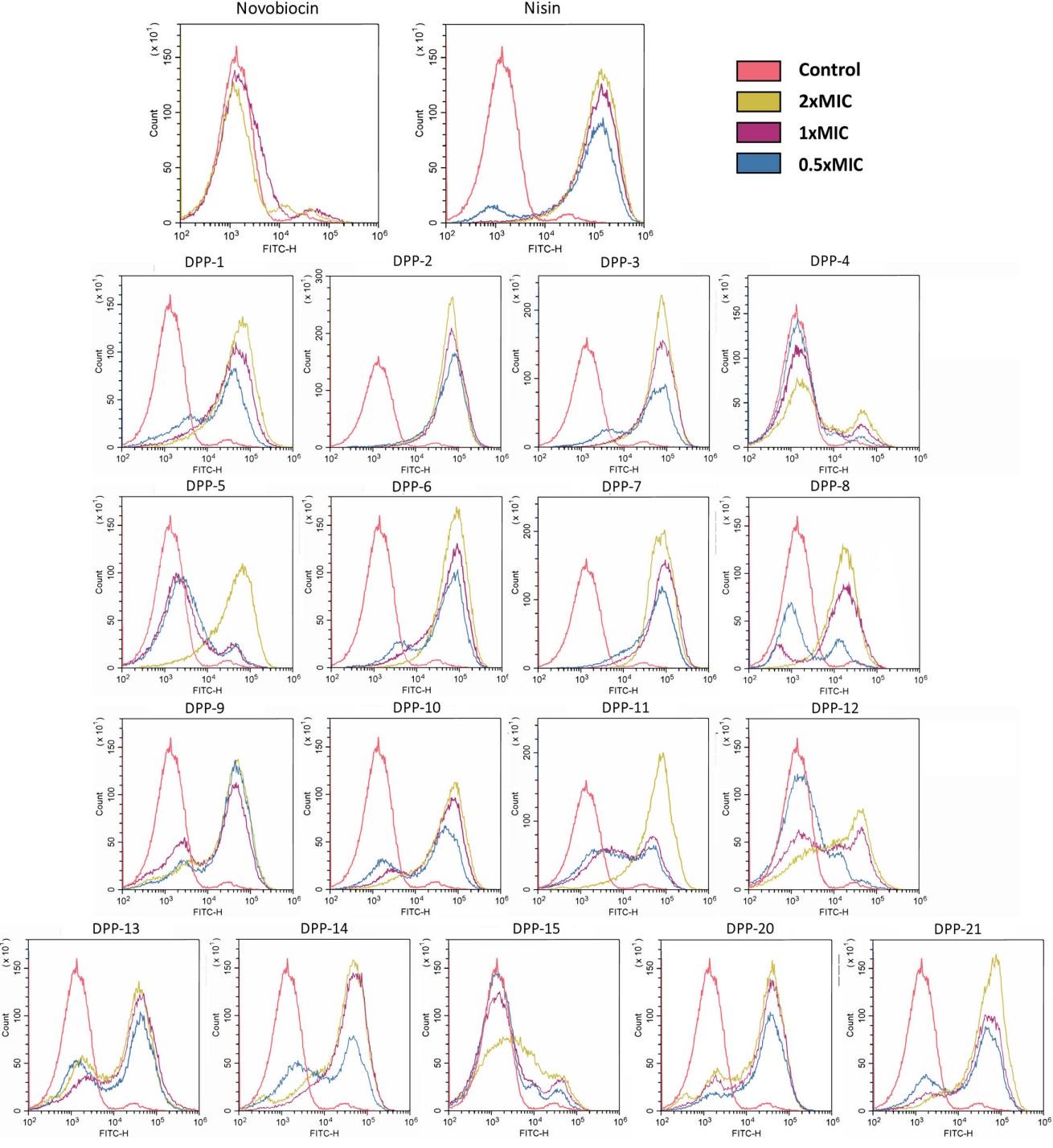

**FIG 5** Permeabilization of *S. aureus* membrane by DPP compounds. *S. aureus* membrane permeabilization was measured using SYTOX-Green fluorescent stain, which can bind to nucleic acid if it penetrates cytoplasmic membranes. Permeabilization is measured as the shift in FITC-H fluorescence compared with the non-treated control (red). DPPs were tested at 0.5× (blue), 1× (purple), and 2× MIC (yellow) with an incubation time of 5 min. Nisin was used as positive control for permeabilization and novobiocin as a negative control.

To test this hypothesis, we performed imaging on HEK293 cells treated with four cytotoxic DPP compounds: **DPP-5** and **DPP-6** (that cause 100% RBC hemolysis) and **DPP-14** and **DPP-20** (not hemolytic). We observed morphological changes when the cells were treated with $LC_{20}$ of hemolytic compounds **DPP-5** and **DPP-6** that resembled membrane blebbing which has been associated to membrane damage in mammalian

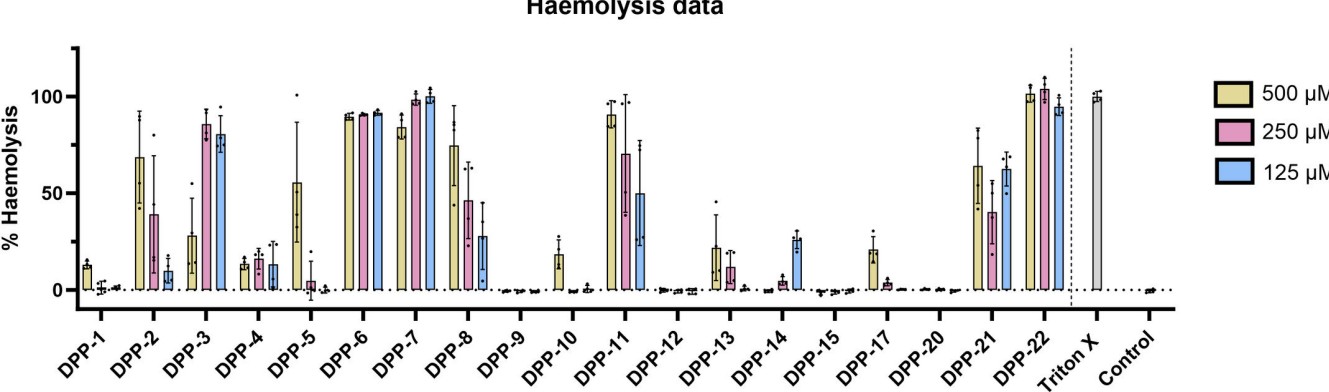

**FIG 6** Hemolysis of sheep red blood cells by DPPs. Percentage of RBC hemolysis compared with Triton-X control is shown after 30 min exposure to 500, 250, and 125 µM of each compound; error bars represent standard deviation ($n = 4$).

cells (Fig. 7C and D; Fig. SF4). At this amplification, no morphological changes were observed with **DPP-14** and **DPP-20**, being comparable with the morphology of the control (Fig. 7B). When cells were observed at higher amplification, we observed a higher number of cells in G2/M phase of the cell cycle when cells were treated with **DPP-14** and **DPP-20** (Fig. 7E and F ; Figu. SF4) (12.24% ± 1.23% and 26.25% ± 4.68% of cells in G2/M phase in **DPP-14** and **DPP-20**, respectively, compared with the 2.68% ± 1.31% in dimethyl-sulfoxide (DMSO) control and 1.63% ± 1.10% in media control), a phenotype that has been reported for Hsp90 inhibitors (42, 43).

## DISCUSSION

Researchers have been trying for over two decades to find inhibitors of bacterial TCSs (44), but no inhibitor has reached clinical evaluation. Here, we build on previous efforts to repurpose mammalian Hsp90 inhibitors targeting the ATP-binding pocket, which has a similar binding fold to the one found in HKs, as TCS inhibitors. We explored the chemical space of the 3,4-diphenylpyrazols, previously reported to inhibit HKs (34). Our collection of DPP analogs was shown to inhibit the growth of Gram-positive but not Gram-negative bacteria, likely due to their capacity to efflux small molecules, and affect accumulation. This was supported by our finding that the MIC of an *E. coli* Δ*tolC* mutant was similar to that of *S. aureus*.

All compounds were shown to fit *in silico* in the ATP pocket of HKs. Docking studies indicate that the substitution pattern on ring A influences the binding mode, with the resorcinol-based derivatives likely to adopt a binding mode analogous to the crystal-bound complex of CCT018159-Hsp90α, an observation also supported by the X-ray crystal structure of resorcinol in complex with the HK CheA (Fig. 4). Compounds with bulky substitutions such as **DPP-8** or that do not contain the resorcinol ring (**DPP-21**) were predicted to have a flipped pose. Six DPPs inhibited HK autophosphorylation *in vitro*. None of the compounds predicted to bind in the flipped orientation were found to inhibit HK autophosphorylation, possibly indicating lower affinity for the ATP pocket. Some compounds that were likely to share a similar binding mode to **DPP-5** did not inhibit autophosphorylation. This is possibly due to precipitation of the compounds affecting the inhibition assays.

One of the main drawbacks of repurposing Hsp90 inhibitors is the potential of retaining toxicity to mammalian cells due to their inhibitory effect on Hsp90 (45). Even though Hsp90 has been identified as an anti-cancer target due to its greater importance in cancer cell protein homeostasis than in normal cells (33), it still plays an important role in normal metabolism (46). Therefore, when repurposing Hsp90 inhibitors, it is important to find chemical spaces that favor binding to HKs while diminishing binding to Hsp90. We saw that DPP compounds still bound with high affinity to Hsp90; thus, more efforts

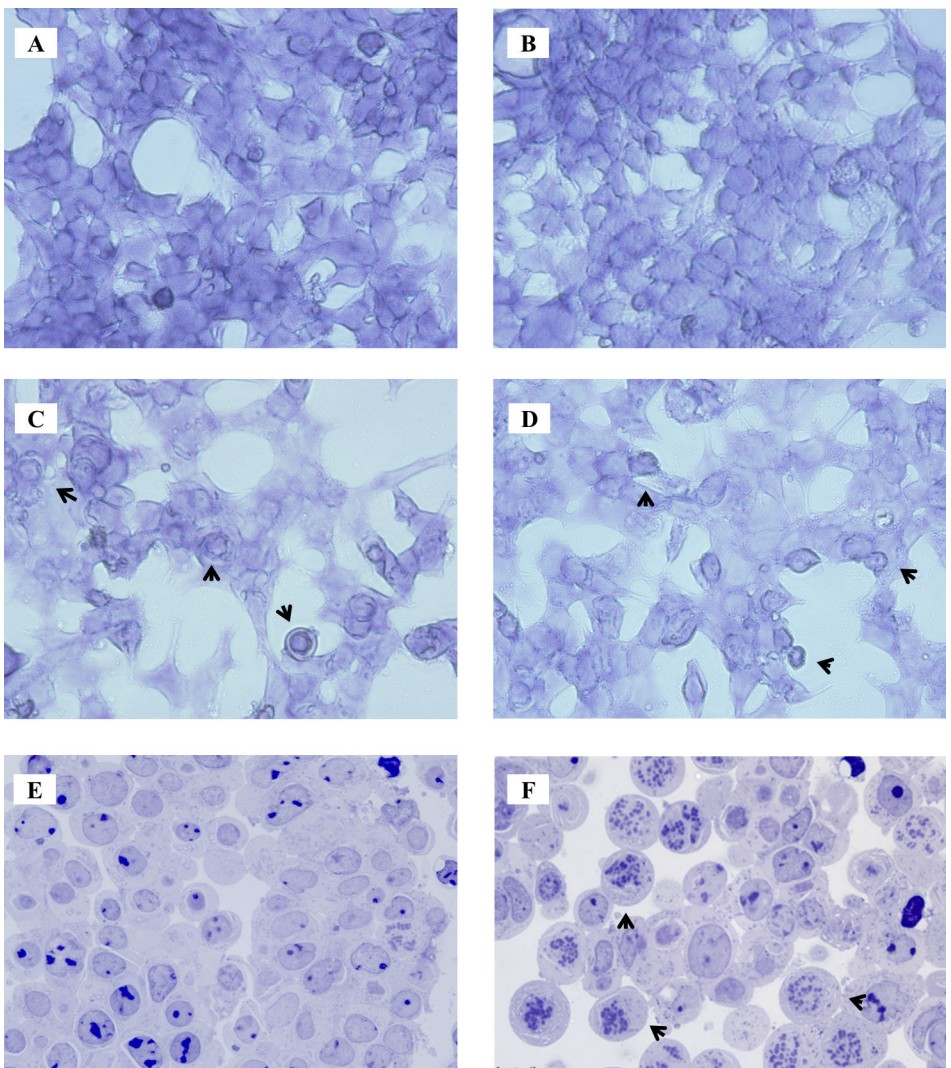

**FIG 7** Cell imaging of HEK293 cells after 24 hours exposure to DPP compounds. (A) HEK293 cells grown in exposure media B. HEK293 cells grown in the presence of LC$_{20}$ concentration of DPP-14 (3 µg/mL) (B), DPP-5 (5µg/mL) (C), or DPP-6 (6 µg/mL) (D) and imaged in 24-well plates at 40× amplification. Exposed cells were also transferred from the plate to a glass slide for visualization at 100× amplification of the exposure media control (E) and cells grown in the presence of LC$_{20}$ (3 µg/mL) DPP-20 (F). Arrows point to morphological abnormalities.

are needed to dissociate Hsp90 activity from HK inhibition. In fact, all compounds presented a high level of toxicity against HEK293 cells (Selectivity Index between 0.45 and 9.14; Table ST3).

The fact that compounds with good antibacterial activity were inactive in the inhibition assay suggests that many of these DPPs inhibit bacterial growth through a mechanism other than HK inhibition. To explore the causality between HK inhibition and antimicrobial activity, we first tested for common off-target effects. Hilliard et al. (47) demonstrated that the TCS inhibitors described in the literature at that time interfered with membrane integrity in *S. aureus* and/or caused hemolysis in mammalian erythrocytes. Most of the tested DPP compounds caused an increase in membrane permeability similar to the nisin control, while only 8 out of the 20 compounds tested caused hemolysis in sheep RBC. Table ST3 shows an overview of all biological parameters of all compounds for SAR analysis. There is no evidence to suggest that disruption of TCS will lead to membrane disruption in *S. aureus* since it can survive in laboratory conditions while lacking all but WalKR TCS (48). The fact that the DPP series inhibit growth of

Gram-negative *E. coli ΔtolC* suggests that WalKR is not the target of our compounds as this TCS is absent in *E. coli*. Thus, we suggest that membrane disruption caused by DPPs is not related to TCS inhibition. There was no correlation between hemolysis or mammalian cell death and membrane-permeabilizing activity in Gram-positive bacteria (Fig. SF1), suggesting different mechanisms of membrane disruption in bacteria and cytotoxicity in mammalian cells.

Even though our data suggests there is one main mechanism of action against bacteria, there seems to be two distinct populations when it comes to cytotoxicity. Less than half of the compounds tested caused hemolysis in RBC. Aberrant cell morphology of HEK293 was observed when treated with two hemolytic compounds (**DPP-5** and **DPP-6**). Observed morphologies resembled membrane blebbing that has been associated to cell membrane damage. The cell morphology of cells treated with two non-hemolytic (**DPP-14** and **DPP-20**) compounds was comparable to that of control cells. However, amplification to 100× shows that both non-hemolytic compounds caused cell-cycle arrest in G2/M phase as shown for some inhibitors of Hsp90. This is not observed for the hemolytic compounds **DPP-5** and **DPP-6** at 100× amplification. Based on these findings, we suggest that cytotoxicity by non-hemolytic DPPs is due to inhibition of Hsp90, further supporting the idea that the mechanism of action against bacteria and mammalian cells is different.

In summary, we further explored the repurposing of the DPP scaffold from inhibition of Hsp90 to inhibition of HKs. Structural analogs were synthesized to explore the chemical space for inhibition of HK autophosphorylation and binding to HKs. Even though in this study, we highlight some of the difficulties in using DPP analogs for antimicrobial discovery, we demonstrated different mechanisms of action for inhibition of bacterial growth and cytotoxicity. Optimization of these compounds toward better physicochemical properties and as potential bacterial membrane disruptors could, together with reducing affinity to Hsp90, lead to good antimicrobials with low toxicity. Finally, the high-resolution crystal structure of the resorcinol fragment can be used as a starting point for the structure-based design of selective HK inhibitors, with compound properties carefully optimized to avoid hydrophobicity, off-target effects, and toxicity in mammalian cells.

## MATERIAL AND METHODS

### Chemical compounds

The chemical compounds **DPP-2**, **4**, **6**, **8**, **9**, **11**, **13**, **21**, **22**, **23**, and **24** were synthesized by the Latvian Institute of Organic Synthesis. Details of the synthesis of the compounds can be found in the study by Solomin et al. (39); the schematic synthesis of **DPP-22** is shown in Fig. S5. Chemical compounds **DPP-1**, **3**, **5**, **7**, **10**, **12**, **14**, **15**, **16**, **27**, **28**, **19**, and **20** were purchased from MolPort (Riga, Latvia). Working stocks of compounds were prepared in DMSO at a concentration of 50 mg/mL.

### Bacterial strains and growth conditions

*Staphylococcus aureus* str. Newman, *Enterococcus faecalis* vanA (strain E0155), *Enterococcus faecium* vanA (strain E1654), *Escherichia coli* ATCC 25922, *Escherichia coli* D21f2 with a truncated LPS barrier, *Escherichia coli* JW5503 (*ΔtolC*) (49), *Pseudomonas aeruginosa* ATCC 27853, and *Pasteurella haemolytica* ATCC 29701 were grown in Mueller-Hinton Broth (MHB) (Oxoid, Basingstoke, UK) and incubated at 37°C.

### Minimal inhibitory concentration

The minimal inhibitory concentration was determined using the microdilution method following the guidelines of the European Committee on Antimicrobial Susceptibility Testing (50). Briefly, a series of twofold dilutions in MHB of each compound were made in a 96-well plate, with final concentrations ranging from 50 to 0.39 µg/mL in 100 µL final volume per well. A hundred microliters of 1:100 dilution of an overnight culture

of the correspondent bacteria in MHB was added to each well. Plates were incubated for 18 hours at 37°C. MIC was recorded as the lowest concentration where no growth was detected as measured by optical density at 600 nm ($OD_{600}$) using Spectramax M5 (Molecular Devices LLC, San Jose, CA, USA). Compounds that did not inhibit growth were re-tested using a higher concentration range (250–1.95 µg/mL). Wells containing bacteria with or without 1% DMSO and medium alone were included as controls in every plate.

## Cell culture and cytotoxicity using Alamar Blue assay

HEK293 cells and the human hepatocellular carcinoma cells (HepG2) were routinely cultured in 75-$cm^2$ culture flasks (Corning Incorporated) on high-glucose Dulbecco's modified Eagle medium (DMEM) containing glutaMax and phenol red (Gibco) supplemented with 1% penicillin/streptomycin (Gibco) and 10% fetal bovine serum (FBS) (Gibco). Cells were maintained at 37°C and 5% $CO_2$.

Cytotoxicity assays with HEK293 and HepG2 cells were performed in 96-well plates seeded with $5 \times 10^4$ cells/well and incubated for 24 hours to reach 80%–90% confluency. Exterior wells were filled with only medium to prevent evaporation. Culture medium was removed from the cells, and 100 µL of exposure medium [high-glucose DMEM without phenol red (Gibco)] was added to avoid a potential interaction between FBS components or antibiotics and tested compounds. A twofold dilution series of compounds in exposure medium (range 100–3.2 µg/mL) was made in a separate 96-well plate, and 100 µL of each of these dilutions was added to the assay plates for 24 hours at 37°C in the presence of 5% $CO_2$. Control wells contained cells with DMEM, DMEM + 1% DMSO (vehicle control), or DMEM + 20% DMSO. Wells without cells were also included as negative control. After exposure, the medium was replaced with 100 µL 10% Alamar Blue (Invitrogen) in exposure medium. After 45 min incubation at 37°C and 5% $CO_2$, fluorescence was measured on a Spectramax M5 plate reader at λex = 541 nm, λem = 590 nm. Cell viability compared with the vehicle control was calculated, and inhibition curves for each compound were fitted (non-linear curve fit, four variables, bottom constrained = 0) in Prism 9 (GraphPad Software, San Diego, USA). The $LC_{50}$ value for 50% cell viability was calculated based on the fitted curves.

## Computational studies

The compounds were prepared for docking using the MOE (38) database wash application to add hydrogens, to generate protonation states and tautomers. The MOE energy minimization application was employed to generate low-energy conformations using the MMFF94x forcefield. The protein with PDB code 1ID0 (51) was imported from the RSCB PDB (52). As implemented in MOE, the protein preparation application was used to add hydrogens, assign bond orders, build missing side chains, and assign protonation states. Only chain A was retained for the docking. Water molecule HOH16 (hereafter defined as W1) was retained, while the remaining water molecules and metals were removed. The prepared compounds were docked using the GOLD-5.2 (53) molecular docking tool. The binding site was defined by the protein atoms within 9 Å from the crystal-bound ligand. W1 was set as fixed (toggle state: on; spin state: fix). ChemPLP was used as the scoring function. The search efficiency was set to very flexible. A protein HBond constraint to the conserved aspartic acid residues was added to penalize poses not forming such an interaction (constraint weight: 10; minimum H-bond geometry weight: 0.005).

## Protein production and purification

Cytoplasmic domains from *S. aureus* PhoR and *E. coli* EnvZ as well the CA domain of *Thermotoga maritima* CheA were cloned in pNIC28-Bsa4 (54) plasmid containing His$_6$-tag expressed in *E. coli* RIL and purified as previously described (21, 55, 56) using His-affinity and size exclusion chromatography. Briefly, *E. coli* RIL strains carrying the

appropriate plasmid were grown in 1 L Luria broth (LB) (Merck Millipore) supplemented with kanamycin (100 µg/mL) with shaking (200 r.p.m.) at 37°C. When $OD_{600}$ reached 0.5, 1 mM isopropyl-ß-D-1-thiogalactopyranoside (IPTG) was added to induce protein expression and cells were grown for 3 more hours. Cells were harvested by centrifugation (4,000 $g$, 4°C), resuspended in 40 mL buffer A (50 mM Tris-HCl pH 8.0, 0.5 M NaCl, 10% glycerol, and 1 mM phenylmethanesulfonyl fluoride), sonicated (4°C, 5 min with pulses of 15 sec at intervals of 1 min), and centrifuged (11,000 $g$, 4°C, 60 min). Supernatant was passed through a 5-mL HisTrap HP column (GE Healthcare) equilibrated in buffer A using the AKTA system (GE Healthcare). His-column was washed with buffer A (80 mL), and a linear gradient of imidazole (range 0 to 0.25 M) in buffer A was applied. The purest fractions evaluated by SDS-PAGE were collected, concentrated by ultrafiltration in a Amicon Ultra-15 (10- or 30-kDa exclusion) (Merck Millipore), and further purified by gel filtration using Superdex 200 column (GE Healthcare), equilibrated in buffer B (50 mM TrisHCl, pH 8.0, 0.2 M NaCl, and 5% glycerol). Purified proteins were then concentrated using ultrafiltration and stored at −80°C.

The Hsp90α C-terminal domain was purified as described by Goode et al. (57). *E. coli* BL21(DE3) expression strains containing GST-Hsp90 N(9-236) plasmid (Addgene: 22481) were grown in 1 L LB media supplemented with 100 µg/mL ampicillin at 25°C and shaking (200 r.p.m.) to $OD_{600}$ 0.5, and protein expression was induced by the addition of 1 mM IPTG, and the culture was grown at 25°C and shaking (200 r.p.m.) overnight. Cells were then pelleted (4,000 $g$, 4°C), resuspended in 40 mL buffer A2 (50 mM Tris-HCl pH 8.0, 0.5 M NaCl, and 1 mM phenylmethanesulfonyl fluoride), sonicated (4°C, 5 min with pulses of 15 sec at intervals of 1 min), and centrifuged (11,000 g, 4°C, 60 min). Supernatant was passed through a 5-mL Glutathione Sepharose 4B (GE Healthcare) equilibrated with buffer A2 using an AKTA system (GE Healthcare). Column was washed using buffer A2, and a linear gradient of glutathione (0–0.25M) in buffer A2 was applied. Fractions containing Hsp90α were collected, mixed, and concentrated by ultrafiltration.

## HK autophosphorylation

Autophosphorylation inhibition assays were performed in kinase assay buffer [100 mM Tris-HCl (pH 8.0), 5 mM $MgCl_2$, 10 mM DTT], containing 5 µg of purified HK and 10 mM ATP for 30 min at 25°C. The reaction was stopped by adding gel loading buffer containing 16% SDS before loading the reaction on a 12% SDS-PAGE. Non-hydrolyzable AMP-PNP (10 mM) was used as control for inhibition of autophosphorylation. In the first screening, inhibitors were added at a concentration 2 mM and compounds giving complete inhibition of autophosphorylation were selected for screening at a range of concentrations (0–1,000 µM). To detect autophosphorylation, the gel was transferred to a nitrocellulose membrane (GE Healthcare) at 100 V in the transfer buffer (25 mM Tris, 192 mM Glycine, 10% Methanol). The membrane was then blocked using Tris-buffered saline solution with 0.1% vol/vol Tween 20 (TBS-T) supplied with 5% bovine serum albumin. The phosphorylated histidine was detected by western blotting by then incubating the membrane for 60 min with 1:1,000 anti-N1-phosphohistidine antibody (Merck Millipore) in TBS-T followed by a washing step for 30 min with TBS-T and a final 60-min incubation with 1:10,000 horseradish peroxide-conjugated anti-rabbit IgG (Promega) in TBS-T. Chemiluminescent reaction was measured using Pierce ECL immunoblotting substrate (Thermo Scientific) in a LAS 3000 image analyzer (Fujifilm Life Science, Tokyo, Japan) followed by densitometric analysis using Multi Gauge image-analysis software (Fujifilm Life Science, Tokyo, Japan). The inhibitor concentrations required to halve the chemiluminescence intensity ($IC_{50}$) were determined using Prism 4.1 (GraphPad Software, San Diego, CA, USA).

## Hsp90 inhibition using fluorescence polarization

Competitive binding of inhibitors to Hsp90 was measured by fluorescence polarization assay as previously reported (58). Inhibitors were assayed in different concentrations (range 0–1,000 µM) in a Nunc black, low binding 384-well plate (Thermo Scientific).

Each reaction was prepared in reaction mix (100 mM Tris-Cl, pH 7.4, 20 mM KCl, 6 mM MgCl$_2$, 2 mM DTT, and 0.1 mg/mL bovine serum albumin) in a final volume of 50 µL containing 350 nM Hsp90 and 100 nM geldanamycin FITC labeled (BPS Bioscience, San Diego, CA, USA) and the inhibitor in the correspondent concentration. Blank control containing no enzyme and no geldanamycin, enzyme-positive control containing no inhibitor, and enzyme-negative control containing no Hsp90 were included. Fluorescence polarization was then measured using Tecan SPARK (Tecan, Grödig, Austria) at wavelengths λex = 485 nm and λem = 530 nm. Binding of FITC-labeled geldanamycin [a well-known Hsp90 inhibitor that fits in the ATP-binding pocket (40)] to Hsp90 results in low fluorescence polarization, and competitive binding of the inhibitors resulted in an increase of fluorescence polarization from free FITC-labeled geldanamycin. IC$_{50}$ values that decrease geldanamycin binding in 50% were calculated using Prism 4.1 (GraphPad Software, San Diego, CA, USA).

## Crystallization and structure determination of the CheA-Resorcinol complex

The CheA-CA domain was crystallized by the sitting-drop vapor-diffusion method at 21°C using 0.4 µL of 25 mg/mL CheA-CA protein mixed with 0.4 µL of reservoir solution containing 30% (wt/vol) PEG 8000, 0.6 M ammonium acetate, 0.065 M sodium acetate pH 4.5. Monoclinic crystals appeared within 24 hours, and 0.2 µL of 20 mM resorcinol solution (Enamine, Frankfurt, Germany) was added to the crystallization drops. After 1 hour, the crystals were harvested, soaked in 40% (wt/vol) PEG 8000 as cryoprotectant, and flash frozen in liquid nitrogen. The crystals were diffracted at XALOC beamline ALBA synchrotron, Barcelona, at 100 K. Programmes within CCP4 suit were used for data processing (59). The structure was solved by molecular replacement using the published structure [PDB 1I58 (60)] as template using PHASER (61) followed by refinement using REFMAC5 (62). Automated model building was performed using BUCCANEER (63) followed by manual model building in COOT. The coordinates and restraints for resorcinol were generated using AceDRG (64). COOT was used again for ligand fitting and chain tracing followed by refinement using REFMAC5. The part of the lid loop corresponding to residues 492–505 could not be built owing to the absence of the corresponding electron density in the area. The data collection and refinement statistics can be found in supplementary data (Table ST4). The coordinates for the solved structure are deposited in the Protein Data Bank with accession code 8PF2.

## Membrane permeability using SYTOX-Green

*S. aureus* str. Newman was grown to exponential phase (OD$_{600}$ = 0.4) in MHB media. Five hundred microliters of bacteria was transferred to 1.5-mL tubes (Eppendorf) and incubated for 5 min with different compounds at concentrations of 0.5× MIC, 1× MIC, and 2× MIC. Nisin (Sigma-Aldrich) and novobiocin (Sigma-Aldrich) were used as positive and negative controls for membrane permeabilization, and stained and unstained untreated controls were also included in each assay. After exposure, samples were centrifuged at 4,000 r.p.m. for 5 min and resuspended in PBS. Bacteria were dyed by adding the fluorescent nuclear stain SYTOX Green (Molecular Probes, Invitrogen) to a final concentration of 0.5 µM and incubating for 5 min in the dark. Samples were centrifuged again and resuspended with PBS to remove unbound stain. Three hundred microliters of each sample was transferred to a clean well in a 96-well clear-bottom plate (Corning Incorporated). Fluorescence was measured in CytoFLEX flow cytometer (Beckman Coulter, Brea, CA, USA) using the green (FITC) channel with the excitation wavelength at 488 nm. Overlay histograms were created with CytExpert Software (Beckman Coulter, Brea, CA, USA).

## Haemolysis assays

The hemolysis assay was adapted from the study by Evans et al. (65). Briefly, sheep RBC 10% washed pooled cells in PBS (Rockland Immunochemicals, Limerick, PA, USA) were

Microbiology Spectrum

diluted 1:5 to give a final concentration of 2% RBCs in PBS. Two hundred microliters of the RBC suspension was transferred to 1.5-mL Eppendorf tubes and incubated with 500, 250, and 125 µM of different compounds, at 37°C and 5% $CO_2$ for 30 min. A non-treated control, 1% DMSO (vehicle control), and complete cell lysis control (Triton-X 2%) were included in each assay. After exposure samples were centrifuged at 4°C at 1,000 $g$ for 15 min and 100 µL of supernatant transferred to a clean well in a 96-well, clear, flat-bottom plate (Corning Incorporated). Released hemoglobin was measured by absorption at 540 nm using a Spectramax M5 plate reader. The experiments were performed in quadruplicate. The percentage of hemolysis was calculated as follows:

$$\text{Hemolysis} = \frac{A_{540\text{nm, testcondition}} - A_{540\text{nm, DMSOcontrol}}}{A_{540\text{nm, TritonXControl}} - A_{540\text{nm, DMSOcontrol}}}$$

## Cell imaging

HEK293 cells were plated in 24-well plates on exposure medium for 24 hours (500,000 cells/well). Cells were exposed to a subset of compounds at a lethal concentration of 20% ($LC_{20}$). After 24 hours exposure, cells were fixed with 2% glutaraldehyde for 15 min, washed with water, permeabilized with 0.5% Triton-X, washed twice with water, and stained with Toluidine blue 0.3%. For 100× amplification, cells were harvested by pipetting up and down in 250 µL of exposure medium. Cells from three wells treated with the same compound were pooled and centrifuged for 5 min at 1,500 r.p.m. The cell pellet was fixed in 0.1 M cacodylate buffer containing 2% glutaraldehyde (vol/vol) (pH 7.4). After 1 hour on ice, the pellet was washed with 0.1 M cacodylate buffer and fixed in 0.1 M cacodylate buffer containing 1% osmium tetroxide (wt/vol). After a further hour on ice, the cell pellet was embedded in hard Epon and 1-µm slides were cut and stained with 1% toluidine blue and 1% borax (wt/vol). Cell abnormalities were assessed by microscopy using a Leica DM6B light microscope (Leica Microsystems B.V., Amsterdam, the Netherlands) at 100× magnification.

## ACKNOWLEDGMENTS

Data collection experiments for the structures reported in the manuscript were carried at XALOC beamline at ALBA synchrotron (Cerdanyola del Valles, Spain) supported by ALBA BAG Proposal 2022075911. We acknowledge the ALBA synchrotron for provision of beam time, and we would like to thank beamline staff for the assistance. This project has received funding from the European Union's Horizon 2020 research and innovation programme under the Marie Sklodowska-Curie grant agreement number 765147 and grants PID2019-108541GB-I00 from the Spanish Government (Ministry of Science and Innovation) and PROMETEO/2020/012 by the Valencian Government to A.M. We also thank the Utrecht Medical Centre (UMC) group from prof. Rob Willems for providing the enterococci.

## AUTHOR AFFILIATIONS

[1]Host-Microbe Interactomics Group, Dept. Animal Sciences, Wageningen University & Research (WUR), Wageningen, the Netherlands
[2]Oxford Drug Design (ODD), Oxford Centre for Innovation, Oxford, United Kingdom
[3]School of Computer Science, University of Buckingham, Buckingham, United Kingdom
[4]Macromolecular Crystallography Group, Instituto de Biomedicina de Valencia-Consejo Superior de Investigaciones Científicas (IBV-CSIC) and CIBER de Enfermedades Raras (CIBERER), Valencia, Spain
[5]Organic Synthesis Methodology Group, Latvian Institute of Organic Synthesis (LIOS), Riga, Latvia

## AUTHOR ORCIDs

Blanca Fernandez-Ciruelos http://orcid.org/0000-0002-7532-2704
Alberto Marina https://orcid.org/0000-0002-1334-5273
Jerry M. Wells http://orcid.org/0000-0002-8743-2652

## FUNDING

| Funder | Grant(s) | Author(s) |
| --- | --- | --- |
| EC \| Horizon Europe \| Excellent Science \| HORIZON EUROPE Marie Sklodowska-Curie Actions (MSCA) | 765147 | Blanca Fernandez-Ciruelos |
| | | Marco Albanese |
| | | Anmol Adhav |
| | | Vitalii Solomin |
| | | Aigars Jirgensons |
| | | Alberto Marina |
| | | Paul W. Finn |
| | | Jerry M. Wells |
| Ministerio de Ciencia e Innovación (MCIN) | PID2019-108541GB-I00 | Alberto Marina |
| GVA \| Conselleria de Cultura, Educación y Ciencia, Generalitat Valenciana (Ministry of Culture, Education and Science of the Generalitat Valenciana) | PROMETEO/2020/012 | Alberto Marina |
| The Netherlands Organization for Health Research and Development (ZonMw) | 50-55100-98-144 | Jerry M. Wells |

## ADDITIONAL FILES

The following material is available online.

### Supplemental Material

**Supplemental material (Spectrum00146-24-S0001.docx).** Fig. S1 to S5; Tables S1 to S4.

### Open Peer Review

**PEER REVIEW HISTORY (review-history.pdf).** An accounting of the reviewer comments and feedback.

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
