## [Reviewer comments · Microbiology Spectrum]

Microbiology Spectrum

Repurposing Hsp90 inhibitors as antimicrobials targeting two-component systems identifies compounds leading to loss of bacterial membrane integrity

Blanca Fernandez-Ciruelos, Marco Albanese, Anmol Adhav, Vitalii Solomin, Arabela Ritchie-Martinez, Femke Taverne, Nadya Velikova, Aigars Jirgensons, Alberto Marina, Paul Finn, and Jerry Wells

Corresponding Author(s): Blanca Fernandez-Ciruelos, Wageningen University & Research

Review Timeline:

Submission Date:	January 16, 2024
Editorial Decision:	February 8, 2024
Revision Received:	April 22, 2024
Accepted:	May 14, 2024

Editor: Brian Conlon

Reviewer(s): The reviewers have opted to remain anonymous.

Transaction Report:

DOI: <https://doi.org/10.1128/spectrum.00146-24>

Re: Spectrum00146-24 (Repurposing Hsp90 inhibitors as antimicrobials targeting two-component systems identifies compounds leading to loss of bacterial membrane integrity)

Dear Dr. Blanca Maria Fernandez-Ciruelos:

Thank you for the privilege of reviewing your work. Below you will find my comments, instructions from the Spectrum editorial office, and the reviewer comments.

Please look at the reviewer comments and try to address their concerns through text modifications. Please also address concerns in your rebuttal.

Revision Guidelines

Sincerely,
Brian Conlon
Editor
Microbiology Spectrum

Reviewer #1 (Comments for the Author):

This paper reports on the synthesis and testing of a set of compounds, based on mammalian HSP90 inhibitors, for their effects on bacteria, especially *S. aureus*. The hypothesis appears to be that such inhibitors might/should have antagonistic activity against kinases of multiple bacterial two-component systems and that such multiple-hits should lead to cell death, most likely

due to membrane disturbance/lysis. The main problem with this approach is that membrane disruption is not a desirable mechanism for antibacterials unless it can be rigorously proven that the lytic activity has no mammalian toxicity. In this case it would be difficult to prove that any lethal outcome was due to an "acceptable" mechanism. In fact (lines 132-135) all the compounds are highly cytotoxic on HEK293 cells while it is stated that the antibacterial and cytotoxic activities are not highly correlated, I think this analysis is incorrect. It is clear that the compounds do not act similarly. Looking at correlations between and among the mammalian cytotoxicity, hemolytic activity, MIC, and bacterial lysis (Figure SF1A) and the associated data, it appears that there may be (at least) two populations; most have similar/correlated effects, but others are outliers. I think the overall low correlations seen are due to this heterogeneity. It would be desirable for all these parameters were shown on a single table so that these outliers could be better delineated and then any SAR could be better shown. Will this lead to any trends indicating that a non-toxic agent is possible? A better discussion/analysis of SAR is needed.

Specific comments:

While it appears that cytotoxicity and hemolysis tests were done in a manner to avoid the effects of serum protein binding (presumed from the explanation that Exposure Medium does not appear to have added FCS), it would still be important to know the extent of serum binding by these many compounds. In the absence of that, MICs in the presence of 50% serum (or equivalent added HSA [43 mg/ml]) could be run.

Reviewer #2 (Comments for the Author):

The authors report a proof-of-concept study repurposing Hsp90 inhibitors as antimicrobials targeting the histidine kinase of bacterial TCSs. Furthermore, the authors eloquently hypothesized that since both the Hsp90 chaperone and bacterial histidine kinase possess a Bergerat fold within the ATP-binding pocket, ATP-competitive Hsp90 inhibitors could be repurposed to inhibit TCS signaling. While the DPP analogs illustrated Gram-positive activity, the activity did not extend to Gram-negative strains tested likely due to efflux pumps. Finally, although the DPP analogs illustrated cytotoxicity through different mechanisms in mammalian cells, the data presented in this manuscript represent a jumping off point to further explore this chemical space and improve specificity to bacterial histidine kinases.

I believe that the authors have presented a very well written manuscript that is scientifically sound as a proof-of-concept for a new antimicrobial target. My major comment is that the authors mainly only cite reviews in regard to TCSs. I would highly recommend citing primary literature to ensure that the researchers that generated the data a review could be formed around are appropriately acknowledged for their hard work. Otherwise, I only have minor comments, a few questions that I felt as a reader were not clear, and a few suggestions for the authors to move forward with this work.

Minor comments

1. Line 33 (and throughout) - Instead of saying mammalian cancer cell lines, its common practice to simply say HEK293 cells or mammalian cell lines. It doesn't need to be noted that it's a cancer cell line unless you elaborate on why its important you're specifically using a cancer cell line.
2. Line 40, I believe should be the plural - antimicrobials
3. Line 43 - change Tis to this
4. Line 46-47 sounds as if you're developing two separate compounds - I would suggest changing the sentence to something along the lines of "...for the development of bacterial TCS inhibitors that act by disrupting bacterial membranes."
5. I would stick to either CCT018159(26) or 3,4-diphenylpyrazole when the compounds are discussed after line 77 as it got confusing when used interchangeably; you can also start using DPP in your introduction rather than waiting until the results section to introduce the acronym
6. Line 153 - why are those specific compounds the most relevant?
7. Table 3 - what does n.t. stand for?
8. Clarify what geldanamycin is used for and why its your comparator as it is not common knowledge. After looking it up, this is a wonderful control but it may confuse the reader if its not explained.
9. Line 236-237 - you state DPP-4, -5, -15 do not cause significant loss of membrane integrity, however DPP-5 is the only compound with efficacy against E.coli, why is this? It would be interesting to see If it disturbs membrane integrity in E. coli
10. In figure 5 there is no data for triton-X control plotted
11. Line 300 - change likely to likely, I also think this sentence is missing a word? Possibly "a similar binding mode [to] DPP-5..."?
12. You mention that DPP-6 at LC20 does not cause any effects to HEK293 cells compared to the control which you attribute to the low concentration, is there a reason you did not increase the concentration to further delineate the reason for toxicity? Did you try this with any other DPP analogs? The one that would be most interesting would be DPP-5 due to its activity against gram-negative and gram-positive strains.
13. Throughout the study you use different HKs from different species (PhoQ, EnvZ, PhoR, Walk, etc) depending on the method you were conducting rather than just focusing on one. Can you elaborate as to why you chose these HKs to evaluate? Also could you elaborate on whether these analogs are likely to work on all TCSs present in a bacterial species? For example S. aureus has 16 (Villanueva, M., García, B., Valle, J. et al. Sensory deprivation in Staphylococcus aureus. Nat Commun 9, 523 (2018)) would you expect that all TCSs that are present are inhibited or is DPP selectively inhibiting specific TCSs?

The authors report a proof-of-concept study repurposing Hsp90 inhibitors as antimicrobials targeting the histidine kinase of bacterial TCSs. Furthermore, the authors eloquently hypothesized that since both the Hsp90 chaperone and bacterial histidine kinase possess a Bergerat fold within the ATP-binding pocket, ATP-competitive Hsp90 inhibitors could be repurposed to inhibit TCS signaling. While the DPP analogs illustrated Gram-positive activity, the activity did not extend to Gram-negative strains tested likely due to efflux pumps. Finally, although the DPP analogs illustrated cytotoxicity through different mechanisms in mammalian cells, the data presented in this manuscript represent a jumping off point to further explore this chemical space and improve specificity to bacterial histidine kinases.

I believe that the authors have presented a very well written manuscript that is scientifically sound as a proof-of-concept for a new antimicrobial target. My major comment is that the authors mainly only cite reviews in regard to TCSs. I would highly recommend citing primary literature to ensure that the researchers that generated the data a review could be formed around are appropriately acknowledged for their hard work. Otherwise, I only have minor comments, a few questions that I felt as a reader were not clear, and a few suggestions for the authors to move forward with this work.

Minor comments

1. Line 33 (and throughout) – Instead of saying mammalian cancer cell lines, its common practice to simply say HEK293 cells or mammalian cell lines. It doesn't need to be noted that it's a cancer cell line unless you elaborate on why its important you're specifically using a cancer cell line.
2. Line 40, I believe should be the plural – antimicrobials
3. Line 43 – change Tis to this
4. Line 46-47 sounds as if you're developing two separate compounds – I would suggest changing the sentence to something along the lines of ..."for the development of bacterial TCS inhibitors that act by disrupting bacterial membranes."
5. I would stick to either CCT018159(26) or 3,4-diphenylpyrazole when the compounds are discussed after line 77 as it got confusing when used interchangeably; you can also start using DPP in your introduction rather than waiting until the results section to introduce the acronym
6. Line 153 – why are those specific compounds the most relevant?
7. Table 3 – what does n.t. stand for?
8. Clarify what geldanamycin is used for and why its your comparator as it is not common knowledge. After looking it up, this is a wonderful control but it may confuse the reader if its not explained.
9. **Line 236-237** – you state DPP-4, -5, -15 do not cause significant loss of membrane integrity, however DPP-5 is the only compound with efficacy against E.coli, why is this? It would be interesting to see If it disturbs membrane integrity in E. coli
10. In figure 5 there is no data for triton-X control plotted
11. Line 300 – change likelty to likely, I also think this sentence is missing a word? Possibly "a similar binding mode [to] DPP-5..."?
12. You mention that DPP-6 at LC20 does not cause any effects to HEK293 cells compared to the control which you attribute to the low concentration, is there a reason you did not increase the concentration to further delineate the reason for toxicity? Did you try this

with any other DPP analogs? The one that would be most interesting would be DPP-5 due to its activity against gram-negative and gram-positive strains.

13. Throughout the study you use different HKs from different species (PhoQ, EnvZ, PhoR, Walk, etc) depending on the method you were conducting rather than just focusing on one. Can you elaborate as to why you chose these HKs to evaluate? Also could you elaborate on whether these analogs are likely to work on all TCSs present in a bacterial species? For example *S. aureus* has 16 (Villanueva, M., García, B., Valle, J. et al. Sensory deprivation in *Staphylococcus aureus*. *Nat Commun* 9, 523 (2018)) would you expect that all TCSs that are present are inhibited or is DPP selectively inhibiting specific TCSs?

Dear Editor and Reviewers,

We thank the reviewers for their constructive comments and suggestions which have helped to improve the quality of the manuscript. Below the reviewers comments are addressed point by point for your consideration. All lines refer to the clean final manuscript.

Reviewer 1

General comments: The hypothesis appears to be that such inhibitors might/should have antagonistic activity against kinases of multiple bacterial two-component systems and that such multiple-hits should lead to cell death, most likely due to membrane disturbance/lysis. The main problem with this approach is that membrane disruption is not a desirable mechanism for antibacterials unless it can be rigorously proven that the lytic activity has no mammalian toxicity. In this case it would be difficult to prove that any lethal outcome was due to an "acceptable" mechanism. In fact (lines 132-135) all the compounds are highly cytotoxic on HEK293 cells while it is stated that the antibacterial and cytotoxic activities are not highly correlated, I think this analysis is incorrect. It is clear that the compounds do not act similarly. Looking at correlations between and among the mammalian cytotoxicity, haemolytic activity, MIC, and bacterial lysis (Figure SF1A) and the associated data, it appears that there may be (at least) two populations; most have similar/correlated effects, but others are outliers. I think the overall low correlations seen are due to this heterogeneity. It would be desirable for all these parameters were shown on a single table so that these outliers could be better delineated and then any SAR could be better shown. Will this lead to any trends indicating that a non-toxic agent is possible? A better discussion/analysis of SAR is needed.

The original idea of this research was to look for novel inhibitors of two-component systems by repurposing inhibitors of the cancer target Hsp90 and further exploring previous research such as the one of Vo *et al.* We indeed hypothesized that antagonistic activity of the compounds against TCS would lead to inhibition of growth or virulence via multiple-hits, however not through membrane damage (for example, it has been reported that deletion of all but WalKR TCS in *S. aureus* does not cause death in rich laboratory media, see Villanueva *et al.*) (Discussion, Lines 269 to 276). Previous research (Hilliard *et al* 1999) showed that a lot of TCS inhibitors were causing inhibition of growth via a non-specific membrane damage mechanism and were never proven to inhibit TCS in bacteria but only *in vitro*. Thus our aim was to determine whether our inhibitors lead to inhibition of growth via a specific two-component system inhibition pathway or through a non-specific mechanism. We found that, while 6 of our compounds showed inhibition of HK *in vitro*, all of them were bactericidal through a mechanism related to bacterial membrane disruption and probably unrelated to TCS inhibition.

We acknowledge that membrane damage is not an ideal mechanism of action due to the possible extension of lytic activity to mammalian cells. However, there are exceptions, e.g. nisin causes pores in bacterial membranes by binding to lipid II, a bacterial component not found in mammalian cells. To see if our compounds were acting by disrupting bacterial membrane in a selective manner we

performed haemolysis and cell imaging assays. We were very interested to find that indeed there are two populations related to cytotoxicity (haemolytic vs. non-haemolytic compounds), while most the compounds showed a similar mode of action against *S. aureus*. We performed new cell imaging for this rebuttal in cells treated with the non-haemolytic DPP-14, and cells treated with the haemolytic DPP-5 and DPP-6. We see that cells treated with DPP-14 show cell cycle arrest phenotype (Figure 7 and Figure SF4), not present in cells treated with DPP-5 and DPP-6. When the cells were not removed from the plate, we saw that DPP-5 and DPP-6 treated cells had morphology disruptions resembling blebbing (which has been related to cell membrane disruption), while cells treated with DPP-14 show no morphological changes when compared to the Control. In our opinion this further strengthens the hypothesis that, at least some compounds cause cytotoxicity and bacterial death through different mechanisms. This information could be used to improve the DPP series of compounds, towards better selectivity as antimicrobials. We have included the new cell imaging (Figure 7 and Figure SF4), and added this information in the Results and Discussion sections (Lines 224 to 234 and Lines 277 to 286). In supplementary Figure SF1, we have also coloured haemolytic compounds in red to show two populations.

Finally, we have included an overview table in Supplementary data (Table ST3) to help the reader in checking the SAR. For better understanding the SAR we have also performed membrane damage, haemolysis of three more compounds (Figure 5 and 6). Unfortunately the SAR for this set of compounds is difficult to analyse. For example, it does look like substitution of the R₁ CF₃ by Me reduces haemolysis, while substitution of X from N to O leads to haemolysis. The substitution of R₃ by O-Me reduces haemolysis while maintaining a similar membrane damage to bacteria when compared to hydroxyl or chlorine substituted compounds (more information about SAR has been included in Lines 203-205 and Lines 209-213).

Specific comment: Cytotoxicity and haemolysis assays have been done in a manner to avoid serum protein binding. It would be important to know the extent of serum binding (i.e. MIC in presence of 50% serum).

In this manuscript we avoided the effect of serum protein binding in our assays so we could study the SAR without external disturbances. We have performed MIC assays in presence of 20% FCS and we see that MIC for all compounds is then >250 µg/ml, indicating that these compounds bind to serum proteins (Lines 127-129). This was expected due to the high cLogP of these compounds (around 3-4). However, improvement to the cLogP of compounds could be possible in further studies to lower the protein binding of these compounds and improve physicochemical properties. It would be specially interesting to grow the resorcinol fragment into novel compounds inhibiting two-component systems or disrupt bacterial membranes, while maintaining good pharmacological properties (Lines 292, 294-296).

In summary, we acknowledge the limitations of the compounds that have been investigated in this manuscript. However, these structures have been studied as potential anti-cancer or antimicrobial agents in the past, so the reported data (cytotoxicity, MIC, mechanism of action and putative inhibition of HKs and Hsp90) could be very valuable in future research on these structures as anti-cancer, membrane disruptors or even two-component system inhibitors.

Reviewer 2

Major comment: Authors mainly only cite reviews in regard to TCS. I would highly recommend citing primary literature to ensure that researchers that generated the data are appropriately acknowledge for their work.

Dear reviewer, thank you for bringing this to our attention. We did indeed review all introduction and change many of the citations to include primary work about two-component systems (see Introduction, citations 1-39).

Minor comments.

Comment 1. This was changed throughout the paper to HEK293 cells or mammalian cells (Line 33, Line 100, Line 123).

Comment 2 and 3. We corrected the typos.

Comment 4. Since we are not sure that membrane disruption occurs through TCS inhibition, I think is correct to refer to inhibitors with these separate activities. The section on Importance has been revised to make this clearer.

Comment 5. We have revised the manuscript, introduced the DPP acronym earlier on. We did not change the use of CCT018159 since it is a compound with specific radicals that possess a DPP core structure. However I included context when mentioned, so the reader does not need to remember the significance of this compound (Lines 77, 79, 84, 95, 99, 102, 153, 187).

Comment 6. We choose to show those compounds because they were the ones that later on showed *in vitro* inhibition against histidine kinases (Table 2). For clarity this information is added on Line 141 and 142.

Comment 7. I changed n.t. (not tested) for the most used n.d. (not done) and include the meaning in the legend.

Comment 8. I included why we use FITC-label geldanamycin in the assay (Line 412) and included a small explanation in the result section (Line 175 and 176).

Comment 9. Indeed compound DPP-5 shows a MIC in *E. coli* of 50 µg/ml. In our experience this is too high to be relevant as precipitation and osmolarity can be confounding factors for the growth

inhibition effect. What we do know is that all DPP compounds show good MIC against *E. coli* lacking TolC efflux pumps. Preliminary data suggests that the compounds lead to disruption of the inner membrane in *E. coli* $\Delta tolC$. However, due to the size of the present manuscript we have decided to add more data on the effect of similar structures against Gram-negative bacteria in a follow-up manuscript.

Comment 10. Triton-X and Control variability is now shown in Figure 6.

Comment 11. Sentence was corrected.

Comment 12. When this experiment was firstly performed we used 3 concentrations (LC_{20} , LC_{50} and LC_{70}). The two highest concentrations led to cell death and no imaging was possible, while LC_{20} did not show morphological changes when using the described protocol in Material and Methods. We agreed that it would be good to show two different cytotoxic populations (see also response to reviewer 1), and we performed the assay again, and image directly in the plate. Changes in morphology now at LC_{20} are obvious for DPP-5 and DPP-6, being similar for both compounds but different from DPP-14 (Figure 7 and Figure SF4 and Lines 224 to 234 and Lines 277 to 286). Some cell blebbing can point to membrane damage or apoptotic response of the cells to the compounds. However, more analysis to confirm this would be necessary. The clear difference seen between DPP-5/DPP-6 and DPP-14, strengthen the idea of two different mechanisms of action against mammalian cells.

Comment 13. Thank you for this question, which raises one of the main “problems to be solved” in the design and development of two-component system inhibitors. In this study we use different HKs for convenience (i.e. PhoR and EnvZ have model ATP pockets and are easily purifiable, PhoQ is the model HK for docking, and the protocol to crystallize CheA is well-established, while the crystallisation of other HKs is still problematic). The ATP-pocket is conserved among TCS, however individual differences, mainly in the lid from this pocket can lead to differences in compound binding. The design of compounds that lead to higher affinity for one TCS could decrease binding to other TCS and vice versa. We do not think that it will be possible to inhibit all TCS in a bacterial species, and what would be the effect of inhibit part or most of them is unknown for most bacteria (except *S. aureus* as stated in Villanueva *et al.*). Some individual TCS are expected to be conditionally essential e.g. during infection of the host so they could be good single HK targets. We are currently working in an opinion paper about TCS inhibition that expands the problems and opportunities of targeting two-component systems, individually and as multi-target.

Re: Spectrum00146-24R1 (Repurposing Hsp90 inhibitors as antimicrobials targeting two-component systems identifies compounds leading to loss of bacterial membrane integrity)

Dear Dr. Blanca Maria Fernandez-Ciruelos:

Your manuscript has been accepted, and I am forwarding it to the ASM production staff for publication. Your paper will first be checked to make sure all elements meet the technical requirements. ASM staff will contact you if anything needs to be revised before copyediting and production can begin. Otherwise, you will be notified when your proofs are ready to be viewed.

Sincerely,
Brian Conlon
Editor
Microbiology Spectrum

Reviewer #1 (Comments for the Author):

Review of spectrum hsp90 paper

This manuscript is improved over the original. It recognizes that there may be 2 mechanisms of cytotoxicity, which is of interest. The study is reasonable and does show inhibition by DPPs of some TCS. That the antibacterial activity of these compounds could not be attributed to that inhibition but rather appeared due to membrane disruption due to other mechanisms is a useful finding. Antibacterial discovery is not easy. It is necessary to tie molecular targeting to the killing mechanism - and this study illustrates the importance of investigating the causality of proposed- target inhibition on killing.

Specific comments

Lines 112-115. This is a run-on sentence. Please break it up into individual sentences.

Line 133. Write out Pasteurella here, as this is the first time it is introduced.

Line 221. May be at least two different mechanisms.

Line 292-294. This is too hopeful a conclusion. Instead of "could," perhaps say "might, if possible," since there is no real support for this.

Line 62 of Supplementary Figs and Table: Supplementary and Percentage are misspelled.

Reviewer #2 (Comments for the Author):

As stated in the first round of revisions the authors report a proof-of-concept study repurposing Hsp90 inhibitors as antimicrobials. Furthermore, the authors thoroughly investigated the limitations, activity, mechanism of action, and cytotoxicity of DPP analogues.

The authors have addressed all of my comments and suggestions to make the manuscript clearer, as well as adding additional supporting data. Finally, I believe that the authors have presented a very well written manuscript that is scientifically sound as a proof-of-concept for a new antimicrobial target.

My only minor comment is a typo:

On line 368 of the marked up manuscript I believe you mean to say 'non-haemolytic' DPP instead of "no-haemolytic"

As stated in the first round of revisions the authors report a proof-of-concept study repurposing Hsp90 inhibitors as antimicrobials. Furthermore, the authors thoroughly investigated the limitations, activity, mechanism of action, and cytotoxicity of DPP analogues.

The authors have addressed all of my comments and suggestions to make the manuscript clearer, as well as adding additional supporting data. Finally, I believe that the authors have presented a very well written manuscript that is scientifically sound as a proof-of-concept for a new antimicrobial target.

My only minor comment is a typo:

On line 368 of the marked up manuscript I believe you mean to say non-haemolytic DPP instead of “no-haemolytic”